 

# Potential dissemination of IncHI2/IncHI2A plasmids carrying *mcr*-9.4 complex transposon in chicken-derived *Enterobacter hormaechei*

Panpan Liu,[1,2] Mengke Ru,[3] Baocheng Hao,[3] Ling Wang,[3] Shengyi Wang,[3] Haipeng Cheng,[4] Dongan Cui[1,2,3]

**ABSTRACT** The escalating global prevalence of antimicrobial resistance(AMR) represents a critical public health challenge, particularly concerning the compromised efficacy of polymyxins—essential therapeutic agents against carbapenem-resistant Gram-negative pathogens. This crisis is exacerbated by the plasmid-mediated horizontal gene transfer mechanism, which facilitates the inter-reservoir dissemination of resistance determinants across anthropogenic, zoogenic, and environmental microbiomes. This study investigated a multidrug-resistant *Enterobacter hormaechei* strain GS32 isolated from a deceased 180-day-old laying hen. Antimicrobial susceptibility testing, whole-genome sequencing, and comparative genomics were employed to analyze resistance profiles, plasmid architecture, and genetic mobility. Conjugation assays assessed plasmid transferability. Results revealed *E. hormaechei* GS32 harbored a 255 kb IncHI2/IncHI2A plasmid carrying *mcr*-9.4(pGS32-1) within a conserved transposon (IS*1R*-*qseB/qseC*-*wbuC*-*mcr*-9.4-IS*903B*) alongside 14 additional resistance genes [e.g., *tet*(D), *mph*(A), and *sul2*] and heavy metal resistance determinants. The pGS32-1 demonstrated high similarity to those in *Salmonella* spp. and *Citrobacter freundii*, suggesting cross-species transmission. Conjugation to EC600 occurred efficiently (frequency: $[7.92 \pm 0.75] \times 10^{-2}$). To our knowledge, the present study provides the first evidence of the presence of an IncHI2 carrying *mcr*-9.4 in *E. hormaechei* isolated from poultry. The pGS32-1 was frequently found in *Enterobacter* sp. (including *E. hormaechei* and *Enterobacter cloacae*), *Salmonella* sp., and other bacteria such as *C. freundii* and *Leclercia adecarboxylata*, indicating the cross-species transmission capability of IncHI2 plasmids, highlighting its role in disseminating polymyxin resistance across ecological niches. These findings underscore the urgent need for enhanced antimicrobial resistance surveillance in livestock and stricter antibiotic stewardship to mitigate the emergence of a multidrug-resistant pathogen under the One Health framework.

**IMPORTANCE** Polymyxin, as the last-line therapeutic agent against carbapenem-resistant Gram-negative bacterial infections, is facing increasing clinical challenges due to the emergence of novel resistance mechanisms. In this study, a strain of *Enterobacter hormaechei* GS32 harboring an IncHI2/IncHI2A-type plasmid (pGS32-1) was isolated from deceased laying hens. This plasmid carries a multidrug resistance gene cluster, including *mcr*-9.4, and exhibits high-efficiency conjugative transfer capability. The *mcr*-9.4 gene is located within a conserved transposon structure (IS*1R*-*qseB/qseC*-*wbuC*-*mcr*-9.4-IS*903B*), colocalized with other resistance genes on the plasmid, suggesting its potential integration as a more complex transposon substructure into this plasmid type. Previous studies have demonstrated that IncHI2-type plasmids are predominantly distributed among *Enterobacteriaceae* species such as *Klebsiella pneumoniae* and *Salmonella* spp. Notably, pGS32-1 exhibits high homology with plasmids identified in *Salmonella* spp.

**Peer Reviewers** Biao Tang, University of the Chinese Academy of Sciences, Hangzhou, China; Piklu Roy Chowdhury, University of Technology Sydney, Sydney, Australia

Address correspondence to Dongan Cui, cda@lzu.edu.cn, or Haipeng Cheng, ccchhhppp@126.com.

The authors declare no conflict of interest.

See the funding table on p. 9.

and *Citrobacter freundii*, indicating the cross-species transmission potential of IncHI2/IncHI2A-type plasmids and their role in expanding the reservoir of resistance genes.

**KEYWORDS** antimicrobial resistance, colistin, *Enterobacter cloacae complex*, *mcr-9*

The *Enterobacter cloacae* complex (ECC) is a common *Enterobacter* species associated with respiratory, urinary tract, and bloodstream infections (1, 2). The rise of numerous multidrug-resistant ECC strains has resulted in a reduced selection of antimicrobial agents for the treatment of these infections (3). *Enterobacter hormaechei*, a member of the ECC, is a significant global pathogen linked to increased morbidity and mortality among hospitalized patients, particularly in carbapenem-resistant cases (4). The escalating prevalence of infections caused by carbapenemase-producing bacteria, coupled with the scarcity of new antimicrobial agents, has positioned polymyxins as a "last-resort" therapeutic option (5, 6). However, the spread of polymyxin resistance genes further restricts available therapies.

Research shows that polymyxin resistance primarily arises from chromosomal mutations in lipopolysaccharide synthesis genes (7). The first plasmid-mediated polymyxin resistance gene, *mcr*-1, was discovered in 2016, and 10 gene variants (*mcr*-1 to *mcr*-10) have since been identified (7, 8). These genes encode a phosphoethanolamine transferase that catalyzes the transfer of phosphoethanolamine to the phosphate group of lipid A, reducing the negative charge of the bacterial outer membrane and its affinity for polymyxin, thereby causing polymyxin resistance. Among *mcr*-like genes, *mcr*-1 and *mcr*-9 are the most prevalent. In 2019, *mcr*-9 was detected in a polymyxin-sensitive clinical isolate of the *Salmonella typhimurium* serotype. Analysis of the strain source of this gene has reported that the *mcr*-9 gene is relatively common in *Enterobacteriaceae*, *Klebsiella*, and *Salmonella* (8, 9). Currently, reports of *mcr*-9-carrying *E. hormaechei* primarily focus on human clinical bloodstream infections (10–14) but also encompass respiratory infections (15, 16), isolates from food sources (17), and feces of healthy chicken flocks (18). Strains carrying only the *mcr*-9 gene, which is the sole *mcr*-like gene, are often colistin-sensitive (9, 19, 20). Global reports on genomic analysis of *mcr*-positive strains indicate that 90.9% of *mcr*-9 strains carry the IncHI2 replicon (21). A high carriage rate of IncHI2 is also observed in *mcr*-1 strains, and it has been reported to coexist with $bla_{NDM-5}$ on the same plasmid (22–24). IncHI2-type plasmids are typically capable of conjugative transfer. Therefore, the potential spread of the *mcr*-9 gene among isolates, coupled with inadequate monitoring, may lead to the emergence of pan-resistant bacteria, thereby posing an unpredictable potential threat to public health.

Herein, we characterized a multidrug-resistant *E. hormaechei* subsp. *hoffmannii* (*E. hormaechei* GS32) isolated from a 180-day-old laying hen. Whole-genome sequencing (WGS) revealed a conjugative IncHI2/IncHI2A plasmid carrying the *mcr*-9.4 polymyxin resistance gene among multiple antimicrobial resistance determinants. Integrative analysis of plasmid architecture and resistance gene organization demonstrated efficient horizontal transfer mechanisms, while phylogenetic tracing of mobile genetic elements uncovered potential dissemination routes for polymyxin resistance in poultry ecosystems.

## RESULTS

### *E. hormaechei* GS32 is a multidrug-resistant strain carrying *mcr-9*

The results of antimicrobial susceptibility testing revealed that *E. hormaechei* GS32 exhibited resistance to multiple antibiotics, including ampicillin, amoxicillin/clavulanic acid, cephalexin, cephalothin, gentamicin, flumequine, florfenicol, tetracycline, and trimethoprim/sulfamethoxazole (Table 1). WGS analysis of *E. hormaechei* GS32 revealed a circular chromosome of 4,597,781 bp with a GC content of 55.34%. Additionally, two plasmids were identified: pGS32-1 (255,043 bp) and pGS32-2 (86,149 bp, no drug resistance gene and complete conjugative transfer module). Resistance gene analysis indicated that only *fosA* was located on the genomic chromosome, whereas all other

**TABLE 1** Detection of antimicrobial susceptibility of *E. hormaechei* GS32

| Antimicrobials | MIC values (mg/L) | | |
|---|---|---|---|
| | *E. hormaechei* GS32 | Transconjugant GS32-EC600 | EC600 |
| Ampicillin | ≥32 (R) | 8 (R) | 8 (R) |
| Amoxicillin/clavulanic acid | ≥32 (R) | 4 (S) | 4 (S) |
| Ticarcillin/clavulanic acid TEC | ≤8 (S) | 16 (S) | 16 (S) |
| Cefalexin | ≥64 (R) | 16 (R) | 16 (R) |
| Cefalotin | ≥64 (R) | 4 (I) | 4 (I) |
| Cefoperazone | ≤4 (S) | ≤4 (S) | ≤4 (S) |
| Ceftiofur | 4 (I) | ≤1 (S) | ≤1 (S) |
| Cefquinome | ≤0.5 (S) | ≤0.5(S) | ≤0.5 (S) |
| Imipenem | ≤0.25 (S) | ≤0.25 (S) | ≤0.25 (S) |
| Gentamicin | ≤1 (*R$^a$) | ≤1 (S) | ≤1 (S) |
| Neomycin | 16 (I) | ≤2 (S) | ≤2 (S) |
| Flumequine | ≥32 (R) | ≥32 (R) | ≥32 (R) |
| Enrofloxacin | 1 (I) | ≤0.5 (S) | ≤0.5 (S) |
| Marbofloxacin | 2 (I) | ≤0.5 (S) | ≤0.5 (S) |
| Tetracycline | ≥16 (R) | ≤1 (S) | ≤1 (S) |
| Polymyxin B | 2 (S) | 2(S) | 0.5(S) |

$^a$*, AES modification.

resistance genes—including *mcr*-9.4, *sul2*, *dfrA12*, *floR*, *qnrS1*, *qepA1*, *aph(3')-la*, *aadA2*, *bleO*, *tet*(D), and *mph*(A)—were situated on pGS32-1. Additionally, pGS32-1 harbored heavy metal resistance genes, such as the *ter* family gene cluster (mediating tellurium resistance), *pcoE* and *pcoS* (mediating copper resistance), and the *RcnRA* efflux system (mediating nickel/cobalt efflux). Given these findings, we subsequently focused on the bioinformatic characteristics of pGS32-1.

## Composition and structure of pGS32-1

The replicon prediction results conducted via PlasmidFinder showed that pGS32-1 harbored two plasmid replicons, IncHI2 and IncHI2A, located in close proximity on the pGS32-1 (Fig. S2). According to VRprofile2 predictions, the *mcr*-9 gene is embedded within a transposon structure spanning from 541 to 6,864 bp (Table S3), followed by consecutive integron structures and an *mph*(A) transposon region. The plasmid also included a fully functional conjugative transfer module consisting of origin of transfer (oriT), relaxase, type IV coupling protein (T4CP), and type IV secretion system (T4SS). Additional predictive analysis performed on the oriTfinder website revealed that the oriT is located between 207,376 and 207,627 bp, the relaxase gene between 202,478 and 205,630 bp, and T4CP between 200,394 and 202,478 bp, and there are two regions associated with T4SS genes. However, subsequent analysis revealed a third region related to T4SS in pGS32-1, approximately located between 123,207 and 132,766 bp of the current sequence (Fig. S4). The Integron Finder tool from Galaxy Pasteur was utilized to predict integrons in pGS32-1 to determine the characteristics of the integron located downstream of *mcr*-9. The results showed that pGS32-1 contained a complex region with two integrons: one containing aadA2 and dfrA12 gene cassettes and the other being a truncated integron (Table S5).

## Comparative genomic analysis of pGS32-1

Blastn alignment results demonstrated that pGS32-1 was frequently found in *Enterobacter* sp. (including *E. hormaechei* and *E. cloacae*), *Salmonella* sp., and other bacteria such as *Citrobacter freundii* and *Leclercia adecarboxylata* (Table S6). Prediction and annotation of plasmid resistance genes revealed that this plasmid harbors multiple types of antimicrobial resistance genes and heavy metal resistance genes. Specifically, the antimicrobial resistance genes identified include *mcr*-9, which confers resistance to

polymyxin; *sul2* and *dfrA12*, which are associated with the folate pathway antagonist class; *floR* for the amphenicol class; *qnrS1* and *qepA1* for the fluoroquinolone class; *aph*(3′)-*Ia* and *aadA2* for the aminoglycoside class; *bleO* for the glycopeptide class; *tet*(D) for the tetracycline class; and *mph*(A) for the macrolide class. The heavy metal resistance genes encompass the ter family gene cluster, which mediates tellurium resistance; *pcoE* and *pcoS*, which are involved in copper resistance; and the *Rcn*RA efflux system that facilitates the nickel/cobalt efflux (Fig. 1).

BLAST alignment revealed a significant sequence similarity between pGS32-1 and other plasmids such as pSW37-267109 and pOYZ4 (with ranges of 77,325–181,865 and 195,203–228,924 bp, respectively). Genomic mapping showed minimal similarity in the 1–77,324 bp region, which contained all antimicrobial resistance genes linked to pGS32-1, marking it as the enrichment zone for these genes. The *mcr*-9 gene was identified within a transposon structure spanning from 541 to 6,863 bp, followed downstream by a continuous integron structure and an *mph*(A) transposon region. A comparative analysis of the *mcr*-9 and its transposon region demonstrated complete coverage and similarity with homologous sequences, suggesting a high degree of conservation for both the *mcr*-9 gene and its transposon structure. Notably, the *mcr*-9 gene in pGS32-1 differed from the known variants *mcr*-9.1, *mcr*-9.2, and *mcr*-9.3 (the 1,439th base was mutated from A to T), but it is identical to the sequence in pCf.1_2 (homology 100%) (accession number: NZ_OK649970), leading to its designation as *mcr*-9.4, following the established nomenclature for *mcr*-9 in pCf.1_2. Both pGS32-1 and pCf.1_2 exhibited a high degree of similarity, sharing not only related genes involved in conjugation transfer and sequences associated with heavy metal resistance but also the same *mcr*-9.4 transposon structure "IS*1R*-*qseB*-like-*qseC*-like-*wbuC*-*mcr*-9.4-IS*903B*"

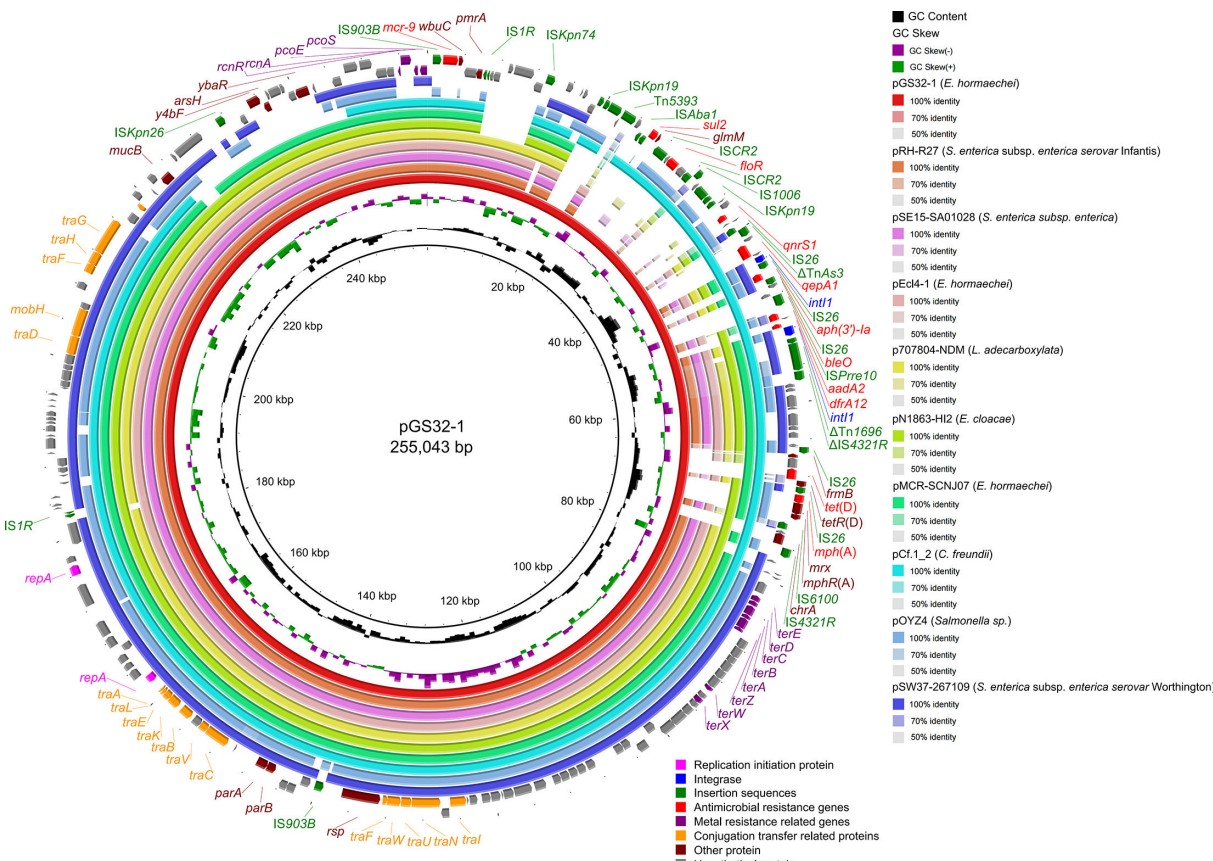

**FIG 1** Comparative analysis of pGS32-1 with pRH-R7(accession no. LN555650), pSE15-SA01028 (accession no. NZ_CP026661), pEcl4-1 (accession no. NZ_CP047741), p707804-NDM (accession no. NZ_MH909331), pN1863-HI2 (accession no. NZ_MF344583), pMCR-SCNJ07 (accession no. NZ_MK933279), pCf.1_2 (accession no. NZ_OK649970), pOYZ4 (accession no. NZ_MN539018), and pSW37-267109 (accession no. NZ_CP051274).

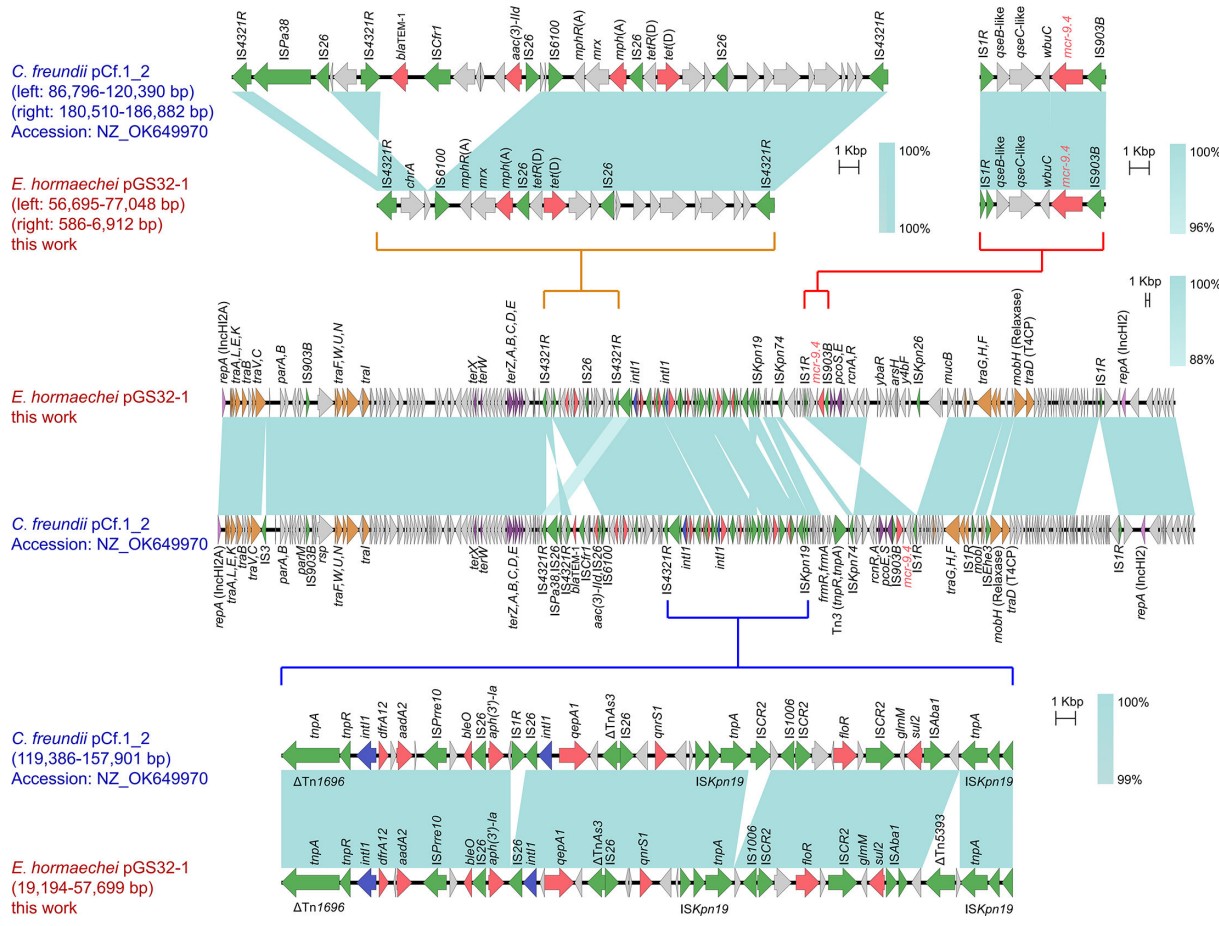

**FIG 2** Comparative genome cluster analysis of pGS32-1.

and relatively similar integron-resistance gene-related regions. Additionally, pGS32-1 contained a *tet*(D) and *mph*(A) transposon structure "IS*6100*-*mphR*(A)-*mrx*-*mphA*-IS*26*-*tetR*(D)-*tet*(D)-*orf*-*orf*-IS*26*" (Fig. 2).

## Conjugative transfer analysis of pGS32-1

PCR detection results confirmed that both the transconjugants and donor strains were positive for the *mcr*-9.4 gene (635 bp) and identified as *Escherichia coli*, matching the recipient strain (Fig. 3). According to the plate counting results, the conjugative transfer frequency was $(7.92 \pm 0.75) \times 10^{-2}$.

## DISCUSSION

As a key member of the highly diverse ECC, *E. hormaechei* is a novel pathogen capable of causing blood infections and bacteremia. Its incidence and mortality rates are considerably high, and it can persist and spread in hospital settings (25). Multiple outbreaks of infection have been reported in hospital ICUs and among neonatal populations (26–28). In recent years, *E. hormaechei* has been isolated from various animals suffering from respiratory diseases and has been reported in animal feces, animal feed, veterinary environments, and food products (29–35). According to a survey analysis, among 118 strains of carbapenem-resistant ECC strains isolated from clinical samples, intestines, and hospital sewage, the detection rate of *E. hormaechei* was significantly higher than that of other strains (53.4%) (36). The emergence of carbapenem-resistant *E. hormaechei* underscores the urgent need for monitoring the spread of polymyxin genes.

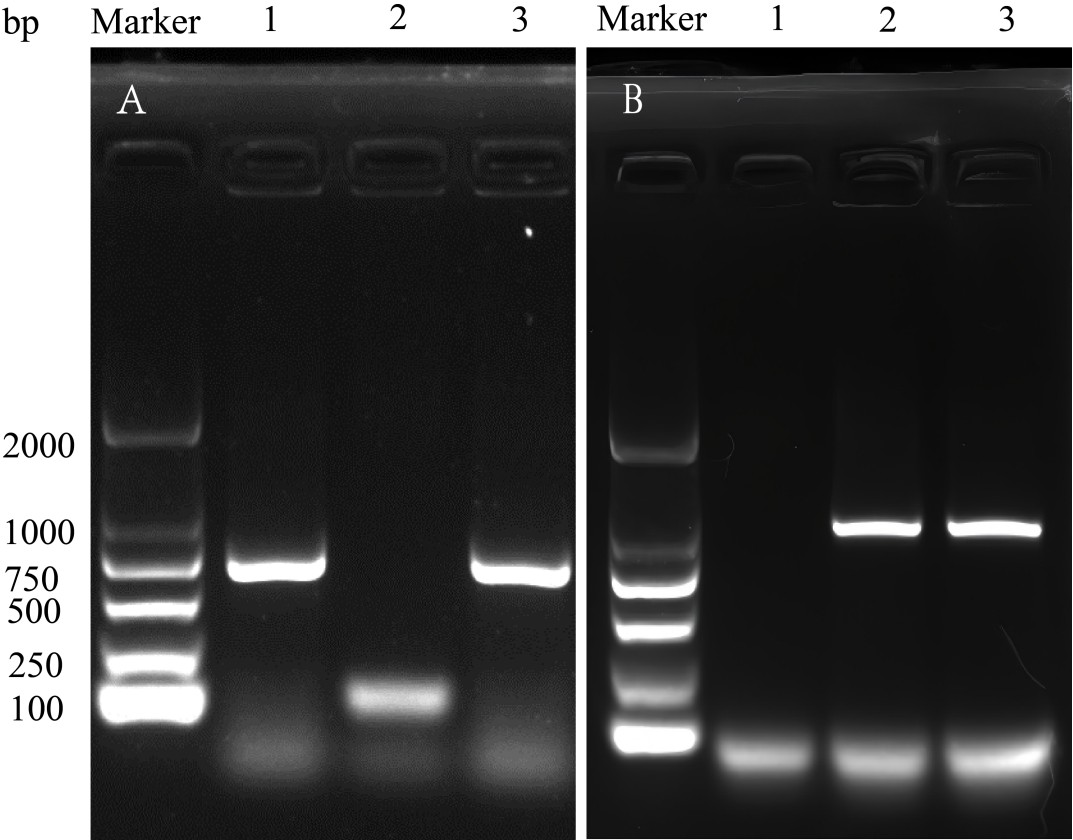

**FIG 3** PCR results for *mcr*-9 (635 bp) (A) and *E. coli*-specific identification (1001 bp) (B) in donor bacteria (*E. hormaechei* GS32), recipient bacteria (EC600), and transconjugants. Lanes are arranged from left to right as follows: Marker, 1 represents donor strain *E. hormaechei* GS32, 2 is recipient strain EC-600, and 3 denotes transconjugants GS32-EC600.

WGS revealed that the *E. hormaechei* GS32 in this study contained an IncHI2-type plasmid carrying *mcr*-9. The IncHI2-type plasmid is one of the most prevalent plasmid types within the *Enterobacteriaceae* family (37). It has a broad host range and is a large (>250 kb) conjugative plasmid capable of mobilizing metal and drug-resistance genes among Gram-negative pathogens (38). A report on the genomic characteristics of *mcr*-positive *E. coli* strains showed a high carriage rate of IncHI2 replicons among *mcr*-9 positive strains (21). Recently, the IncHI2 plasmid harboring *mcr*-9 has been identified as a "super-plasmid" associated with antimicrobial resistance genes (39). The observed conjugation frequency ($[7.92 \pm 0.75] \times 10^{-2}$) combined with broad host range compatibility (*Salmonella* spp., *C. freundii*) validates these plasmids' role as antimicrobial resistance (AMR) transmission vectors across ecological niches.

Comparative genomic analysis showed that heavy metal resistance genes and conjugative transfer-related modules were the conserved sequences of pGS32-1. Multidrug resistance genes are carried by insertion sequences, transposable elements, or integrons, forming a multidrug resistance gene enrichment region and showing significant variability within the plasmid. The integron-resistance gene-associated region of pGS32-1 harbors two adjacent integrons. The first integron carries the *dfrA12* and *aadA2* genes, with its 3′-conserved segment (3′-CS) replaced by IS*Prre10*, an insertion element belonging to the ISCR family. Additionally, the *bleO* and *aph* (3′)-*la* resistance genes are located in close proximity to IS*Prre10*. The second integron, as predicted by Galaxy, does not contain a resistance gene cassette and is a truncated integron. However, upstream of this integron are the *qepA1* and *qnrS1* genes, which mediate fluoroquinolone resistance. Furthermore, this region encompasses an IS*Kpn19* composite transposon that may facilitate the intracellular transfer of the *floR* and *sul2* genes.

Compared with IncHI2 reference plasmid R478, pGS32-1 has a similar skeleton structure, but there is an insertion variable region from IS*4321R* to IS*903B* (40). An incomplete Tn*1696*-like structure (ΔTn*1696*) was identified downstream of IS*4321R*, and no *mer* resistance gene cluster was detected on the plasmid. It is hypothesized that this resulted from the replacement of the original integron 3′-CS by the downstream IS*Kpn19* composite transposon. A truncated 17 bp IR*tnp* was observed on one side of the *tnpA* gene in ΔTn*1696* (compared to the 38 bp IR*tnp* in Tn*1696*), which is consistent with the truncation of IR*tnp* caused by IS*4321* targeting the IR of Tn*1696* in plasmid pK29 (41). Notably, an identical truncated IR*tnp* sequence was also present adjacent to the first upstream IS4321, although no Tn*1696* transposase was identified nearby. It is speculated that the insertion of the *mphA* and *tet*(D) region may have been accompanied by complex homologous recombination events involving IS*4321R*.

The pCf.1-2 is highly similar to pGS32-1 and has the same *mcr*-9.4 transposon structure: "IS*1R*-*qseB*-like-*qseC*-like-*wbuC*-*mcr*-9.4-IS*903B*." Studies have shown that *qseC* genes encode histidine kinases and *qseB* genes encode homologous response regulatory proteins, which together constitute a two-component regulatory system. Colistin at subinhibitory concentration (4 µg/mL) can activate the QseC-QseB regulatory system located downstream of *mcr*-9, induce *mcr*-9 expression, and significantly increase MIC (42). Numerous studies have shown that isolates harboring *mcr*-9 are often sensitive to polymyxin. Additional evidence indicates that the role of *mcr*-9 in conferring polymyxin resistance in *E. coli* is considerably less pronounced than that of *mcr*-1, and subinhibitory levels of polymyxin can significantly elevate the transcription levels of *mcr*-9, *qseB*, and *qseC*. This implies that *mcr*-9 can potentially spread among different bacterial populations without demonstrating resistance and could play a role in polymyxin resistance under appropriate conditions. Therefore, it is imperative to enhance global surveillance of the prevalence of *mcr*-9 in humans and animals.

## Conclusion

The present study reports the first IncHI2/IncHI2A (pGS32-1) carrying *mcr*-9.4 in poultry-derived *E. hormaechei*. The plasmid harbors a conserved transposon (IS*1R*-*qseB*/*qseC*-*wbuC*-*mcr*-9.4-IS*903B*), 14 resistance genes [e.g., *tet*(D) and *mph*(A)], and functional conjugative modules, enabling efficient horizontal transfer (frequency: (7.92 ± 0.75) × 10$^{-2}$). Genomic comparisons revealed high similarity to plasmids in *Salmonella* and *Citrobacter*, indicating cross-species transmission risks. The coexistence of *mcr*-9.4 with inducible QseB/QseC regulators highlights potential undetected polymyxin resistance under routine surveillance. These findings underscore the One Health threat posed by IncHI2 plasmids in livestock ecosystems. Urgent actions include enhanced AMR monitoring in agriculture, stricter antibiotic stewardship, and research into plasmid-host-environment interactions to curb pan-resistant pathogen emergence.

## MATERIALS AND METHODS

### Isolate identification and antimicrobial susceptibility testing

The test strain *E. hormaechei* GS32 was isolated from a deceased 180-day-old laying hen in Gansu. In a laminar flow hood, the chicken's skin was cut open to expose the abdominal cavity. Approximately 1 g of liver tissue was excised using sterilized scissors, minced in an Eppendorf tube, and homogenized in 10 times the volume of phosphate-buffered saline. The supernatant from the homogenized tissue was then spread onto a brain heart infusion plate. Its identification was confirmed via 16S ribosomal RNA sequencing utilizing a pair of universal primers: 27f (5′-AGAGTTTGATCA TGGCTCAG-3′) and 1492r (5′-TAGGGTTACCTTGTTACGACTT-3′). Analyze the susceptibility to 16 antimicrobial agents, using the VITEK 2 COMPACT system (bioMérieux, France) and AST-GN 96 cards (batch number 6862595203, the concentration of antimicrobial listed in ile FS1), following the manufacturer's instructions.

## Genome sequencing and assembly

The genome of *E. hormaechei* GS32 was sequenced using a Pacbio Sequel II and DNBSEQ platform at the Beijing Genomics Institute (Shenzhen, China). The PacBio platform employed four SMRT cells equipped with Zero-Mode Waveguide arrays to generate subreads, with those shorter than 1 kb excluded. Canu software was used for data self-correction. Draft genomic unitigs, which represent unambiguous clusters of fragments, were assembled using the high-quality corrected circular consensus sequence subreads generated by Canu. Single-base corrections were performed using GATK (https://www.broadinstitute.org/gatk/) to enhance the precision of the genome sequences. Subsequently, the genome was annotated using Diamond software and the National Center for Biotechnology Information Prokaryotic Genome Annotation Pipeline.

## Bioinformatics

Antimicrobial resistance genes were identified using ResFinder 4.1.1 (https://genepi.food.dtu.dk/resfinder) with a 96% similarity threshold. Virulence factors were detected via VFDB_VFanalyzer (https://www.mgc.ac.cn/cgi-bin/VFs/v5/main.cgi?func=VFanalyzer). Genomic islands, prophages, and integrative conjugative elements were predicted using VFprofile2 (https://tool2-mml.sjtu.edu.cn/VRprofile/home.php). Plasmid replication initiator types were determined by PlasmidFinder 2.1.2 (https://cge.food.dtu.dk/services/PlasmidFinder/), and plasmid transfer capabilities were evaluated using oriTfinder (https://bioinfo-mml.sjtu.edu.cn/oriTfinder/). Integrons were predicted using the Integron Finder tool on Galaxy Pasteur (https://galaxy.pasteur.fr/). The BLAST Ring Image Generator software was utilized to generate circular comparison images of the raw and reference plasmids, while Easyfig 2.0 software was employed to create linear comparison diagrams of multiple genomic loci surrounding *mcr*-9.4 and other resistance genes.

## Conjugation assays

*E. hormaechei* GS32 carrying *mcr*-9 (donor strain) was conjugated with rifampicin-resistant EC600 (recipient strain) through plasmid conjugation experiments following the methodology of Wang et al. (43). The donor and recipient bacterial strains were combined in a 1:3 volume ratio during their logarithmic growth phase ($OD_{600} = 0.5$) and incubated at 37°C with shaking at 180 rpm for 24 hours. Transconjugants were subjected to selection on Luria-Bertani agar plates supplemented with rifampicin (100 mg/L) and polymyxin B (1 mg/L) to identify those with reduced sensitivity to polymyxin B. Meanwhile, rifampicin-resistant plates were prepared to enumerate the total number of recipient and transconjugants. The presence of *mcr*-9.4 genes in the conjugants was confirmed by PCR (*mcr*-9F: 5′-GCGGTTGAAAGGCGTATGT-3′, *mcr*-9R: 5′-CAAATCGCGGTC AGGATTAT-3′), and antimicrobial susceptibility testing was conducted on the transconjugants to verify the successful transfer of plasmids harboring these target genes.

## ACKNOWLEDGMENTS

This work was supported by the Gansu Province Major Science and Technology Project (24ZD13NA008) and Talent Scientific Fund of Lanzhou University.

Some assistance with bioinformatics was supplied by Shanghai Winnerbio Technology Co., Ltd. (Shanghai, China).

P.L. and M.R.: methodology, investigation, validation, and writing—review and editing. B.H. and L.W.: data curation and writing—review and editing. S.W.: writing—review and editing. H.C. and D.C.: conceptualization; project administration; writing—review and editing; and funding acquisition. All authors have reviewed and approved the final version of the manuscript.

## AUTHOR AFFILIATIONS

[1]State Key Laboratory of Animal Disease Control and Prevention, College of Veterinary Medicine, Lanzhou Veterinary Research Institute, Lanzhou University, Chinese Academy of Agricultural Sciences, Lanzhou, China

[2]Gansu Province Research Center for Basic Disciplines of Pathogen Biology, Lanzhou, China

[3]Lanzhou Institute of Husbandry and Pharmaceutical Sciences, Chinese Academy of Agriculture Sciences, Lanzhou, China

[4]Beijing Kangmu Biological Technology Co., LTD, Beijng, China

## AUTHOR ORCIDs

Panpan Liu  http://orcid.org/0000-0001-8715-8879
Haipeng Cheng  http://orcid.org/0009-0007-8723-4510
Dongan Cui  http://orcid.org/0000-0003-1073-2814

## FUNDING

| Funder | Grant(s) | Author(s) |
| --- | --- | --- |
| Gansu Province Major Science and Technology Project | 24ZD13NA008 | Dongan Cui |
| Talent Scientific Fund of Lanzhou University | | Dongan Cui |

## AUTHOR CONTRIBUTIONS

Panpan Liu, Investigation, Methodology, Validation, Writing – review and editing | Mengke Ru, Investigation, Methodology, Validation, Writing – review and editing | Baocheng Hao, Data curation, Writing – review and editing | Ling Wang, Data curation, Writing – review and editing | Shengyi Wang, Data curation, Writing – review and editing | Haipeng Cheng, Conceptualization, Funding acquisition, Project administration, Writing – review and editing | Dongan Cui, Conceptualization, Funding acquisition, Project administration, Writing – review and editing

## DATA AVAILABILITY

Nucleotide sequences were submitted to the GenBank database under accession numbers CP189870–CP189872 for the *E. hormaechei* GS32 genome and its associated plasmids.

## ETHICS APPROVAL

All applicable international, national, and/or institutional guidelines for the care and use of animals were followed.

## ADDITIONAL FILES

The following material is available online.

### Supplemental Material

**Supplemental material (Spectrum01979-25-S0001.docx).** Files S1 to S6.

### Open Peer Review

**PEER REVIEW HISTORY (review-history.pdf).** An accounting of the reviewer comments and feedback.

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
