## [Reviewer comments · Microbiology Spectrum]

Microbiology Spectrum

Potential Dissemination of IncHI2/IncHI2A Plasmids Carrying *mcr-9.4* Complex Transposon in Chicken-derived *Enterobacter Hormaechei*

Panpan Liu, Mengke Ru, Baocheng Hao, Ling Wang, Sheng Wang, Haipeng Cheng, and Dongan Cui

Corresponding Author(s): Dongan Cui, Lanzhou University

Review Timeline:

Submission Date:	July 2, 2025
Editorial Decision:	August 19, 2025
Revision Received:	October 17, 2025
Accepted:	November 24, 2025

Editor: Catherine Logue

Reviewer(s): Disclosure of reviewer identity is with reference to reviewer comments included in decision letter(s). The following individuals involved in review of your submission have agreed to reveal their identity: Biao Tang (Reviewer #1); Piklu Roy Chowdhury (Reviewer #2)

Transaction Report:

DOI: <https://doi.org/10.1128/spectrum.01979-25>

Re: Spectrum01979-25 (Potential Dissemination of IncHI2/IncHI2A Plasmids Carrying *mcr-9.4* Complex Transposon in Chicken-derived *Enterobacter Hormaechei*)

Dear Prof. Dongan Cui:

Thank you for the privilege of reviewing your work. Below you will find my comments, instructions from the Spectrum editorial office, and the reviewer comments.

Revision Guidelines

Sincerely,
Catherine Logue
Editor
Microbiology Spectrum

Reviewer #1 (Comments for the Author):

This study reports an avian *Enterobacter hormaechei* strain carrying the IncHI2/IncHI2A type "super plasmid" (pGS32-1), which carries a novel polymyxin resistance gene *mcr-9.4* and a cluster of multidrug resistance genes, and confirms its efficient conjugation and transfer ability. The research data is detailed and the methods are standardized. The results are of great value for understanding the transmission mechanism of ARGs among animals, the environment and humans, and meet the

requirements of drug resistance monitoring under the One Health framework. The overall logic of the article is clear, but some parts need further supplementation or adjustment to enhance scientific rigor.

1. Line25, "antimicrobial resistance (AMR)" should be deleted "(AMR)". Line 28-29, "horizontal gene transfer (HGT)" should be deleted "(HGT)"; Line 46, it is recommended that no abbreviations in abstract, such as AMR.
2. Line31-32, "Enterobacter hormaechei strain (E. hormaechei GS32)". It is suggested to write it as "Enterobacter hormaechei strain GS32";
3. Line38 line120, "tet(D)" should be written as "tet(D)". Please check the correct way to write the drug resistance gene in the full text.
4. Line42, "β-lactams" should be written as "β-lactams".
5. Line43, "superplasmid" needs to be clearly defined in manuscript.
6. Line48, "pan-resistant pathogens" should be used with caution as the manuscript has not confirmed that the strains are resistant to all antibiotics.
7. Line 86, "Among mcr-like genes, mcr-1 and mcr-9 are the most prevalent." Gene should be in italics.
8. Line96-97 and line111, "mcr-1 strains, and it has been reported to coexist with blaNDM-5", "mcr-9.4", Please write the ARGs in the correct format.
9. Line 96-98: Relevant literature is recommended for citation. DOI: 10.1016/j.jgar.2022.06.002 , DOI: 10.1007/s12275-022-1597-y , DOI: 10.1128/spectrum.01257-22
10. Line116, "4,597,781", spaces between numbers and punctuation marks, it should as "4, 597, 781".
11. Line118, "FosA" should be written as "fosA".
12. In Table 1, the MIC value of "GM" (gentamicin) is marked as "{less than or equal to}1 (*R)", but the MIC of EC600 is "<1 (S)". It is necessary to explain the criteria for determining drug resistance (such as CLSI or EUCAST breakpoints). In addition, what does the "*" represent?
13. Line134 and 139 and 144 and 155,"he genome (S 2)", "(Figure 1, S 3)", "(S 4)", "(S 5)", "(S 6)". It should be written as "Table S2, Table S3, Table S4, Table S5, Table S6".
14. FIG 1, FIG 2, FIG 2, It should be written as "Figure 1", "Figure 2", "Figure 3".
15. The legend in Figure 2 is written incorrectly. Please correct it and check the full text. It should be pGS32-1.
16. Line157, "drug resistance genes", it is suggested to be written as "antimicrobial resistance genes".
17. Line 175, "antibiotic resistance genes", it is suggested to be written as "antimicrobial resistance genes".
18. Line189, "pS.GS32-1", it is suggested to be written as "pGS32-1".
19. Line181-184, "Notably, the mcr-9 gene in pGS32-1 differed from the known variants mcr-9.1, mcr-9.2, and mcr-9.3, but matched the sequence in pCf.1_2 (Accession number: NZ_OK649970), leading to its designation as mcr-9.4, following the established nomenclature for mcr-9 in pCf.1_2." This sentence mentions the differences between mcr-9.4 and mcr-9.1, mcr-9.2, mcr-9.3, but does not describe in detail where the differences lie. Furthermore, what is the homology and consistency between the mcr-9.4 gene and the mcr-9.4 gene in plasmid pCf.1_2?
20. Line252, "pGS32-1harbors" should be written as "pGS32-1 harbors".
21. Line 260, "pS.GS32-1", please check if the writing is correct.
22. The number of repetitions and standard deviations of the conjugation assays should be supplemented to verify the reliability of the transfer frequency.

Reviewer #2 (Comments for the Author):

The manuscript by Liu et al. mainly reports the identification of an IncHI2 plasmid carrying the mcr-9.4 transposon in an *Enterobacter hormaechei* isolated from the liver of a deceased chicken. The authors used an integrative approach combining genome sequence analysis, antibiotic resistance phenotype profiling, and conjugation assay analysis to show that the mcr-9.4 gene is located within a transposon akin to one recently described in the *Citrobacter*-associated plasmid, pCf . 1_2. Although the approach and analysis seem thorough, I have a few reservations about this report.

Firstly, the choice of counter-selection used in the conjugation assay needs clarification. The authors mention that 1 mg/l colistin was used for counter-selecting transconjugants in the methods section, which aligns with the gene of interest. However, throughout the manuscript, colistin is not mentioned. Table 1 lists polymyxin B, but colistin is actually polymyxin E. Furthermore, there are several references to "polymyxin" and "PB" or "Polymyxin B" in the manuscript, but none to polymyxin E. Please clarify.

Secondly, genome assembly reveals the presence of two plasmids within the genome. I understand that pGS32-1, which carries the mcr-9.4 gene and related transposon, is the primary focus of this manuscript. However, the existence of a second plasmid introduces considerable ambiguity into the narrative and is captured in Table 1. The table shows that the tetracycline resistance profile of the transconjugant matches that of the recipient, not the donor. With the transfer of pGS32, I would expect tetracycline resistance to also transfer to the recipient since the tet genes are on pGS32-1. It's plausible that during the conjugation assay, the transposon carrying the mcr-9.4 gene transposed to the second plasmid and was transferred to the recipient *E. coli*. This scenario is entirely possible, considering that the counter-selection in the conjugation assay relied solely on a phenotype conferred by the mcr-9.4 gene, not other genes on the plasmid. Transconjugants with the second plasmid, containing a copy of the transposed mcr-9.4 gene, could have been selected, which may challenge the authors' claims about the conjugation

experiment. Alternatively, there may be two or three distinct subpopulations, each harbouring different combinations of the plasmids and *mcr-9.4* gene, that were selected in the assay, thereby skewing the reported conjugation frequency. Neither the counter-selection method nor the confirmatory PCR data rule out these possibilities. Therefore, I suggest: (1) that the authors provide more details about the second plasmid, especially its ability to be mobilised; (2) that they use different antibiotics to select for pGS32-1, such as streptomycin, trimethoprim, or macrolides, given that the plasmid also carries other resistance genes; and (3) that the confirmatory PCR targets genes on the plasmid backbone rather than any gene capable of lateral movement between different replicons within the cell.

Finally, I do not see enough data to confirm the presence of two different integron classes, specifically *In0*. It is clear that two integrons exist, but identifying *In0* requires more detailed analysis. I recommend describing it as a complex region with two integrons: one containing *aadA2* and *dfrA12* gene cassettes, and the other being a truncated integron. If the authors wish to delve into details of integron classes, I suggest they refer to the original 1996 paper by Stokes and Hall, which describes *In0*, *In2*, and *In5* and details characteristics that define *In0* (<https://pmc.ncbi.nlm.nih.gov/articles/PMC178208/pdf/1784429.pdf>).

The manuscript also needs careful editing before resubmission, as there are quite a few sentences with grammatical errors.

Minor comments:

1. Line 43: I am unsure why a 255kb plasmid is called a "superplasmid." Literature includes examples of several *IncHI2* plasmids larger than 255kb, so I doubt the adjective relates to size. If it aims to create an analogy to plasmid *pCf.1_2*, then the authors should first demonstrate homology. Overall, I find the term "superplasmid" unnecessary.
2. Lines 87-88: Remove the repetition within the same sentence - "polymyxin-sensitive" and "sensitive to polymyxin".
3. Lines 133-134: "...located in close proximity in the genome (*S2*).". I believe the authors mean that the two replicase genes were on the same unitig, described here as plasmid pGS32-1.
4. Please indicate the name of the "compound" used in reference number 39.
5. Line 350- 351: I am not sure what the authors mean by "receptor organisms and splicers" in relation to the conjugation assay. Please rephrase to use standard terminologies used to describe donor/recipient and transconjugants.

Reviewer #3 (Comments for the Author):

This manuscript seeks to describe an *HI2* plasmid found in an *E. hormacheae* from a dead chicken. A genome sequence for the isolate was determined and it carries a putative colistin resistance gene *mcr9.4* genes that does not appear to confer significant resistance along with other several relevant resistance genes (Table 1). The analysis of the plasmid is not adequately connected to the large literature on plasmids of this type dating back decades and hence lacks informed depth. It uses raw outputs from various databases or search tools. This is not appropriate. These outputs are a first round and need to be carefully examined in the light of what is known. For example, the identity of the genes in the *HI2* backbone are well known and repeating this information adds nothing to the literature.

Conjugation is claimed selecting for a small increase in resistance to colistin. However, the transconjugant is not tetracycline and hence the plasmid cannot be in it. So, this claim is not supported by the data presented.

Please consult the literature on compound/composite transposons. They never end with two different IS. Hence these claims are not supported by the data presented. A review by Partridge et al. doi: 10.1128/CMR.00088-17 may help in this regard. This must be corrected throughout.

Specific comments.

1. *HI2* plasmids are large but they are not "super plasmids". Remove this term.
2. In Table 1, for clarity use the antibiotic names and remove the abbreviations.
3. Remove Fig. 1 - it may assist your analysis but is of no value in a publication.
4. Beware the mistakes in databases. In Fig. 3, *TnAs1*, 2, 3 do not exist. They are the transposition modules of *Tn21*, *Tn1696* etc. *ISVs3* is not an IS it represents part of *CR2* also known as *ISCR2*. Relabel correctly.
4. *HI2* plasmids only transfer under very specific conditions specifically low temperature. See Gilmour et al. *Plasmid* 2004 52: 182-202, and Cain & Hall *JAC* 2012 67:1121-1127.
5. The most important feature of a resistance region is the boundary between the plasmid backbone and incoming DNA. Where is the resistance region(s) located. The backbone can be found easily using the references in 4 above.
6. The transconjugants must be tested for resistance to all of the antibiotics predicted to be present from the sequence.
7. The Discussion is too long.

Dear Editor and Reviewers,

On behalf of my co-authors, we thank you very much for your consideration of our work. We deeply appreciate the reviewer's positive and constructive comments and suggestions regarding our manuscript, titled entitled "**Potential Dissemination of IncHI2/IncHI2A Plasmids Carrying *mcr-9.4* Complex Transposon In Chicken-derived *Enterobacter Hormaechei***" (Spectrum01979-25). Thank you for giving me the opportunity to revise, so as to improve the level of the article. We have studied the reviewer's comments carefully and have revised relevant parts. We have also responded point by point to each of the reviewer's comments, as listed below.

Thank you and best regards,

Yours sincerely,

Reviewer #1 (Comments for the Author):

This study reports an avian *Enterobacter hormaechei* strain carrying the IncHI2/IncHI2A type "super plasmid" (pGS32-1), which carries a novel polymyxin resistance gene *mcr*-9.4 and a cluster of multidrug resistance genes, and confirms its efficient conjugation and transfer ability. The research data is detailed and the methods are standardized. The results are of great value for understanding the transmission mechanism of ARGs among animals, the environment and humans, and meet the requirements of drug resistance monitoring under the One Health framework. The overall logic of the article is clear, but some parts need further supplementation or adjustment to enhance scientific rigor.

Thank you very much for your recognition of the manuscript and our work.

Responses to Reviewer Comments

1. Line25, "antimicrobial resistance (AMR)" should be deleted "(AMR)". Line 28-29, "horizontal gene transfer (HGT)" should be deleted "(HGT)"; Line 46, it is recommended that no abbreviations in abstract, such as AMR.
2. Line31-32, "*Enterobacter hormaechei* strain (*E. hormaechei* GS32)". It is suggested to write it as "*Enterobacter hormaechei* strain GS32";
3. Line38 line120, "tet(D)" should be written as "tet(D)". Please check the correct way to write the drug resistance gene in the full text.
4. Line42, "β-lactams" should be written as "β-lactams".

Response:

We thank the reviewer's comment. We fully agree with your 1-4 suggestions and have revised them in the manuscript.

5. Line43, "superplasmid" needs to be clearly defined in manuscript.

Response:

We thank the reviewer's comment. We have reviewed and revised the improper use of the super plasmid statement in the full text.

6. Line48, "pan-resistant pathogens" should be used with caution as the manuscript has not confirmed that the strains are resistant to all antibiotics.

Response:

We thank the reviewer's comment. For the improper use of 'pan-resistant pathogens', we have modified it to 'multidrug-resistant pathogens'.

7. Line 86, "Among *mcr*-like genes, *mcr*-1 and *mcr*-9 are the most prevalent." Gene should be in italics.
8. Line96-97 and line111, "*mcr*-1 strains, and it has been reported to coexist with *bla*_{NDM-5}", "*mcr*-9.4", Please write the ARGs in the correct format.

Response:

We thank the reviewer's comment. We have corrected this kind of problem in full manuscript.

9. Line 96-98: Relevant literature is recommended for citation. DOI: 10.1016/j.jgar.2022.06.002, DOI: 10.1007/s12275-022-1597-y, DOI: 10.1128/spectrum.01257-22

Response:

We thank the reviewer's comment. The new literature you provide is exactly what we need, which makes the article content professional and rich.

10. Line 116, "4,597,781", spaces between numbers and punctuation marks, it should be "4, 597, 781".

Response:

We thank the reviewer's comment. We have corrected this kind of problem in full manuscript.

11. Line 118, "FosA" should be written as "fosA".

Response:

We thank the reviewer's comment. We have corrected the corresponding position in the manuscript.

12. In Table 1, the MIC value of "GM" (gentamicin) is marked as " ≤ 1 (*R)", but the MIC of EC600 is " < 1 (S)". It is necessary to explain the criteria for determining drug resistance (such as CLSI or EUCAST breakpoints). In addition, what does the "*" represent?

Response:

Dear reviewer, thank you for your rigorous and responsible review of this manuscript. As you reminded, we verified the whole table, which is a writing error. In addition, the MIC results of Transconjugant GS32-EC600 and EC600 are missing '=', which has been corrected. '*' represents AES revision. We have annotated below Table 1. Advanced Expert System TM (AES) is a software program used to confirm and interpret the drug sensitivity results of VITEK® 2 Systems. After the biological confirmation function is enabled, AES can automatically check the drug sensitivity results when the VITEK® 2 System analysis program processes the drug sensitivity results. AES can be used to further analyze drug sensitivity (AST) results to confirm the results and discover drug-resistant phenotypes. AES analysis was based on the MIC interpretation guidelines (CLSI).

13. Line 134 and 139 and 144 and 155, "the genome (S 2)", "(Figure 1, S 3)", "(S 4)", "(S 5)", "(S 6)". It should be written as "Table S2, Table S3, Table S4, Table S5, Table S6".

14. FIG 1, FIG 2, FIG 2, It should be written as "Figure 1", "Figure 2", "Figure 3".

Response:

We thank the reviewer's comment. We have corrected the corresponding position in the

manuscript.

15. The legend in Figure 2 is written incorrectly. Please correct it and check the full text. It should be pGS32-1.

Response:

We thank the reviewer's comment. We correct Figure 2 and unify the manuscript as pGS32-1.

16. Line157, "drug resistance genes", it is suggested to be written as "antimicrobial resistance genes".

Response:

We thank the reviewer's comment. We have corrected this kind of problem in full text.

17. Line 175, "antibiotic resistance genes", it is suggested to be written as "antimicrobial resistance genes".

Response:

We thank the reviewer's comment. We have corrected this kind of problem in full text.

18. Line189, "pS.GS32-1", it is suggested to be written as "pGS32-1".

Response:

We thank the reviewer's comment. The manuscript is unified as pGS32-1.

19. Line181-184, "Notably, the *mcr-9* gene in pGS32-1 differed from the known variants *mcr-9.1*, *mcr-9.2*, and *mcr-9.3*, but matched the sequence in pCf.1_2 (Accession number: NZ_OK649970), leading to its designation as *mcr-9.4*, following the established nomenclature for *mcr-9* in pCf.1_2." This sentence mentions the differences between *mcr-9.4* and *mcr-9.1*, *mcr-9.2*, *mcr-9.3*, but does not describe in detail where the differences lie. Furthermore, what is the homology and consistency between the *mcr-9.4* gene and the *mcr-9.4* gene in plasmid pCf.1_2?

Response:

We thank the reviewer's comment. The sequence of the *mcr-9* gene in pGS32-1 was identical to that of *mcr-9* in pCf.1 _ 2. We added homology results to the manuscript and modified the inaccurate expression.

20. Line252, "pGS32-1harbors" should be written as "pGS32-1 harbors".

Response:

We thank the reviewer's comment. We have corrected the corresponding position in the manuscript.

21. Line 260, "pS.GS32-1", please check if the writing is correct.

Response:

We thank the reviewer's comment. We have corrected this kind of problem in full manuscript.

22. The number of repetitions and standard deviations of the conjugation assays should be supplemented to verify the reliability of the transfer frequency.

Response:

We thank the reviewer's comment. We calculate the results and show the results in the form of standard deviation in the manuscript.

Reviewer #2 (Comments for the Author):

The manuscript by Liu et al. mainly reports the identification of an IncHI2 plasmid carrying the *mcr-9.4* transposon in an *Enterobacter hormaechei* isolated from the liver of a deceased chicken. The authors used an integrative approach combining genome sequence analysis, antibiotic resistance phenotype profiling, and conjugation assay analysis to show that the *mcr-9.4* gene is located within a transposon akin to one recently described in the Citrobacter-associated plasmid, pCf . 1_2. Although the approach and analysis seem thorough, I have a few reservations about this report.

Firstly, the choice of counter-selection used in the conjugation assay needs clarification. The authors mention that 1 mg/l colistin was used for counter-selecting transconjugants in the methods section, which aligns with the gene of interest. However, throughout the manuscript, colistin is not mentioned. Table 1 lists polymyxin B, but colistin is actually polymyxin E. Furthermore, there are several references to "polymyxin" and "PB" or "Polymyxin B" in the manuscript, but none to polymyxin E. Please clarify.

Response:

Thank you very much to the reviewers for pointing out and giving advice on this issue. We have corrected the problem in the whole manuscript.

Secondly, genome assembly reveals the presence of two plasmids within the genome. I understand that pGS32-1, which carries the *mcr-9.4* gene and related transposon, is the primary focus of this manuscript. However, the existence of a second plasmid introduces considerable ambiguity into the narrative and is captured in Table 1. The table shows that the tetracycline resistance profile of the transconjugant matches that of the recipient, not the donor. With the transfer of pGS32, I would expect tetracycline resistance to also transfer to the recipient since the tet genes are on pGS32-1. It's plausible that during the conjugation assay, the transposon carrying the *mcr-9.4* gene transposed to the second plasmid and was transferred to the recipient *E. coli*. This scenario is entirely possible, considering that the counter-selection in the conjugation assay relied solely on a phenotype conferred by the *mcr-9.4* gene, not other genes on the plasmid. Transconjugants with the second plasmid, containing a copy of the transposed *mcr-9.4* gene, could have been selected, which may challenge the authors' claims about the conjugation experiment. Alternatively, there may be two or three distinct subpopulations, each harbouring different combinations of the plasmids and *mcr-9.4* gene, that were selected in the assay, thereby skewing the reported conjugation frequency. Neither the counter-selection method nor the confirmatory PCR data rule out these possibilities. Therefore, I suggest: (1) that the authors provide more details about the second plasmid, especially its ability to be mobilised; (2) that they use different antibiotics to select for pGS32-1, such as streptomycin, trimethoprim, or macrolides, given that the plasmid also carries other resistance genes; and (3) that the confirmatory PCR targets genes on the plasmid backbone rather than any gene capable of lateral movement between different replicons within the cell.

Response:

Thank you very much for the reviewer 's logical reasoning. For your proposal, we agree that the mobility of the second plasmid (pGS32-2) is a key factor affecting the efficiency of conjugative transfer. Therefore, we analyzed the mobility of pGS32-2 and found that it did not have a complete conjugative transfer module(This information has been supplemented in the section ' *E. hormaechei* GS32 is a multidrug-resistant strain carrying *mcr-9*'), so the deviation of conjugative transfer frequency could be excluded. As far as we know, the insertion sequence and the transposon can theoretically enter the recipient bacteria through conjugation, transformation, and transduction. The transformation requires the recipient bacteria to be in a state of competence, and we did not perform this treatment. The transduction also requires the specific phage of the recipient bacteria to complete, and our experiments are performed in the biosafety cabinet for strict aseptic operation. These conditions support the reliability of our experimental results. Unfortunately, the corresponding author of this study is no longer able to retrieve the relevant experimental materials from the original institution (Address 3). We sincerely apologize for not being able to conduct additional tests to further enrich the data presented in this study.

oriT & Relaxase	T4SS	T4CP	Auxiliary protein	Cargo genes	Summary
Type				Location	Locus tag/Gene name
oriT region				-	-
Relaxase				70607..75613 (+)	orf103
T4CP				68334..70607 (+)	orf102
Auxiliary protein				-	-
T4SS gene cluster				16127..33561	orf21; orf23; orf25; orf27; orf28; orf30; orf31; orf32; orf33; orf34; orf37
ARG				-	-
VF				-	-
Metal resistance				-	-
Degradation				-	-
Symbiosis				-	-
Anti-CRISPR				-	-

Finally, I do not see enough data to confirm the presence of two different integron classes, specifically In0. It is clear that two integrons exist, but identifying In0 requires more detailed analysis. I recommend describing it as a complex region with two integrons: one containing *aadA2* and *dfrA12* gene cassettes, and the other being a truncated integron. If the authors wish to delve into details of integron classes, I suggest they refer to the original 1996 paper by Stokes and Hall, which describes In0, In2, and In5 and details characteristics that define In0 (<https://pmc.ncbi.nlm.nih.gov/articles/PMC178208/pdf/1784429.pdf>).

Response:

Thanks for your proposal, our description is likely to cause the same confusion to readers. We must clarify to you that the In0 of the manuscript has nothing to do with the standard In0 you gave in this article. The results of the integron prediction table provided in this study were using the IntegronFinder 2.0 tool of Galaxy Pasteur (<https://galaxy.pasteur.fr/>). The tool will judge the results as complete, In0 or CALIN according to the integrity of the integron, where In0 represents that the integron contains integrase, but no gene cassette is found, that is, the integron is incomplete, that is, what you call 'a truncated integron '. Therefore, we very much agree with your proposal and have changed the relevant expressions in the manuscript.

The manuscript also needs careful editing before resubmission, as there are quite a few sentences with grammatical errors.

Response:

Thank you very much for the reviewer 's points and proposals on this issue. We have revised the relevant expressions in the manuscript for this issue.

Minor comments:

1. Line 43: I am unsure why a 255kb plasmid is called a "superplasmid." Literature includes examples of several IncHI2 plasmids larger than 255kb, so I doubt the adjective relates to size. If it aims to create an analogy to plasmid pCf.1_2, then the authors should first demonstrate homology. Overall, I find the term "superplasmid" unnecessary.

Response:

Thank you very much for the reviewer 's points and proposals on this issue. We have deleted the relevant expressions in the manuscript for this issue.

2. Lines 87-88: Remove the repetition within the same sentence - "polymyxin-sensitive" and "sensitive to polymyxin".

Response:

We thank the reviewer's comment. We have corrected the corresponding position in the manuscript.

3. Lines 133-134: "...located in close proximity in the genome (S2)." I believe the authors mean that the two replicase genes were on the same unitig, described here as plasmid pGS32-1.

Response:

Thank you very much for the reviewer 's points and proposals on this issue. We have revised the relevant expressions in the manuscript for this issue.

4. Please indicate the name of the "compound" used in reference number 39.

Response:

Thank you very much for the reviewer 's points and proposals on this issue. We reviewed the original literature and corrected it to ' colistin '.

5. Line 350- 351: I am not sure what the authors mean by "receptor organisms and splicers" in relation to the conjugation assay. Please rephrase to use standard terminologies used to describe donor/recipient and transconjugants.

Response:

We thank the reviewer's comment. We have corrected this kind of problem in full manuscript.

Reviewer #3 (Comments for the Author):

This manuscript seeks to describe an HI2 plasmid found in an *E. hormacheae* from a dead chicken. A genome sequence for the isolate was determined and it carries a putative colistin resistance gene *mcr-9.4* genes that does not appear to confer significant resistance along with other several relevant resistance genes (Table 1). The analysis of the plasmid is not adequately connected to the large literature on plasmids of this type dating back decades and hence lacks informed depth. It uses raw outputs from various databases or search tools. This is not appropriate. These outputs are a first round and need to be carefully examined in the light of what is known. For example, the identity of the genes in the HI2 backbone are well known and repeating this information adds nothing to the literature.

Conjugation is claimed selecting for a small increase in resistance to colistin. However, the transconjugant is not tetracycline and hence the plasmid cannot be in it. So, this claim is not supported by the data presented.

Please consult the literature on compound/composite transposons. They never end with two different IS. Hence these claims are not supported by the data presented. A review by Partridge et al. doi: 10.1128/CMR.00088-17 may help in this regard. This must be corrected throughout.

Response:

Thank you very much for your questions in the comments and the relevant literature provided. According to the literature provided by you, we have further expanded the analysis of plasmids, revised the sequence-related naming of plasmid gene clusters, and elaborated and discussed the analysis results in the discussion, which will provide readers with greater reading value.

Specific comments.

1. HI2 plasmids are large but they are not "super plasmids". Remove this term.

Response:

Thank you very much for the reviewer's points and proposals on this issue. We have deleted the relevant expressions in the manuscript for this issue.

2. In Table 1, for clarity use the antibiotic names and remove the abbreviations.

Response:

We thank the reviewer's comment. It has been modified as proposed.

3. Remove Fig. 1- it may assist your analysis but is of no value in a publication.

Response:

We agree with your proposal and delete Figure 1.

4. Beware the mistakes in databases. In Fig. 3, TnAs1, 2, 3 do not exist. They are the transposition modules of Tn21, Tn1696 etc. ISVsa3 is not an IS it represents part of CR2 also known as ISCR2.

Relabel correctly. 4. HI2 plasmids only transfer under very specific conditions specifically low temperature. See Gilmour et al. Plasmid 2004 52: 182-202, and Cain & Hall JAC 2012 67:1121-1127.

Q1、 In Fig. 3, TnAs1, 2, 3 do not exist. They are the transposition modules of Tn21, Tn1696 etc.

Response:

Regarding the first issue, the nucleotide sequences originally annotated as TnAs2 and TnAs3 in plasmid pEh32-1 of the sequenced strain GS32 were extracted and compared with Tn21 and Tn1696. The results indicated that the region containing TnAs2 exhibited high similarity to the transposase and resolvase of Tn21 and Tn1696, particularly Tn1696. We appreciate the reviewer's suggestion, and the label for TnAs2 in the figure has been revised to Tn1696 accordingly. However, the alignment of the TnAs3 sequence revealed no significant similarity to Tn21 or Tn1696. This may be attributed to the fact that this segment does not reside within a transposase region. Furthermore, annotation via ISfinder indicated that this sequence shares the highest similarity with TnAs3. Therefore, we request to retain the original designation, TnAs3.

qseqid	sseqid	pident	length	mismatch	gapopen	qstart	qend	sstart	send	evalue	bitscore
TnAs2	AF071	82.797	2982	489	21	1	2973	3000	34	0	2643
-tnpA	413.3										
TnAs2	AF071	77.84	537	113	6	13	546	3551	3018	1.98E-	327
-tnpR	413.3									92	
TnAs2	U1233	100	2973	0	0	1	2973	3006	34	0	5491
-tnpA	8.3										
TnAs2	U1233	100	558	0	0	1	558	3566	3009	0	1031
-tnpR	8.3										

* The sequences for *TnAs2-tnpA* and *TnAs2-tnpR* were retrieved from the corresponding genes of plasmid pGS32-1 in the sequenced strain S. GS32.

* AF071413.3 represents Tn21, and U12338.3 represents Tn1696.

* No alignment results were obtained for the sequences extracted for TnAs3.

Query= TnAs3-tnpA

Length=942

Sequences producing significant alignments	IS Family	Group	Origin	Score (bits)	E. value
TnAs3	Tn3		Aeromonas salmonicida	1126	0.0
IS15DIV	IS6		Salmonella typhimurium	145	2e-34
IS15DII	IS6		Salmonella panama	145	2e-34
IS15DI	IS6		Salmonella panama	145	2e-34
IS15	IS6		Salmonella panama	145	2e-34
IS26	IS6		Proteus vulgaris	145	2e-34
ISSen15	IS6		Salmonella enterica	54.0	4e-07
ISVsa3	ISCR		Vibrio salmonicida	42.1	0.002
ISCaa16	IS6		Candidatus Amoebophilus	38.2	0.027
ISKpn19	ISKra4	ISAZba1	Klebsiella pneumoniae	36.2	0.11
ISAZba1	ISKra4	ISAZba1	Azospirillum brasilense	36.2	0.11
ISAbas2	IS6		Acinetobacter baumannii	34.2	0.42
ISEfa15	IS21		Enterococcus faecium	34.2	0.42

Q2、 ISVsa3 is not an IS it represents part of CR2 also known as ISCR2.

Response:

Thank you for your review, according to your proposal, we have been modified in the figure.

Q3、 HI2 plasmids only transfer under very specific conditions specifically low temperature. See Gilmour et al. *Plasmid* 2004 52: 182-202, and Cain & Hall *JAC* 2012 67:1121-1127.

Response:

Dear Reviewer, your question is a very noteworthy direction. According to the relevant literature (Maher D, Taylor DE. Host range and transfer efficiency of incompatibility group HI plasmids. *Can J Microbiol.* 1993 Jun;39(6):581-7. doi: 10.1139/m93-084.), we found that the HI2 plasmid can undergo conjugation transfer under the conditions of this study, and the high frequency conjugation frequency can be completed under low temperature conditions. In this study, there were two plasmids in the GS32 isolate. Plasmid 2 did not have a complete conjugative transfer module and did not contain drug resistance genes. The possible effect of plasmid 2 on the conjugation efficiency of pGS32-1 was excluded. It was verified by PCR that the IncHI2 plasmid pGS32-1 we focused on did undergo conjugative transfer.

5. The most important feature of a resistance region is the boundary between the plasmid backbone and incoming DNA. Where is the resistance region(s) located. The backbone can be found easily using the references in 4 above.

Response:

We thank the reviewer for their valuable suggestions. Based on the methodology described by Gilmour et al. (*Plasmid* 2004; 52:182–202), we compared the genomic composition of plasmid pGS32-1 from this study with that of the *Serratia marcescens* plasmid R478 (accession no. BX664015.1). The results indicate that the two plasmids share a similar backbone structure. However, the region of pGS32-1 spanning from IS4321R to IS903B could not be aligned with R478, confirming this segment as an inserted variable region. These findings are consistent with our previous analysis.

Based on "Cain & Hall *JAC* 2012 67:1121–1127", the Tn1696-like structure within the sequence was re-identified; however, a complete Tn1696 (U12338.3) was not detected. Specifically, the original Tn1696 *tnpA* and *tnpR* genes were identified adjacent to the integrin in pGS32-1, but the integrons carried by these two elements were different. Moreover, no *mer* resistance gene cluster was identified in pGS32-1. This may be attributed to the replacement of the original integrin's 3'-CS (3'-conserved segment) by the downstream ISKpn19 composite transposon. Additionally, a 17 bp IR_{tnp} was identified on one flank of ΔTn1696 *tnpA*, which was notably truncated compared

to the 38 bp IR_{tmp} of Tn1696. According to the study by Cain & Hall (JAC 2012 67:1121–1127), a similar truncation of IR_{tmp} was observed in pK29, resulting from the targeting of Tn1696 IR by IS4321, which is consistent with the presence of an IS4321R insertion flanking ΔTn1696 *tmpA* in our sample. Notably, the same truncated IR_{tmp} sequence was identified adjacent to the first upstream IS4321, although no Tn1696 transposase was detected nearby. It is hypothesized that the insertion of the *mphA* and *tet(D)* region may have been accompanied by complex homologous recombination involving IS4321R. Accordingly, the comparative gene cluster diagram was revised.

6. The transconjugants must be tested for resistance to all of the antibiotics predicted to be present from the sequence.

Response:

Your proposal to test the changes in antibiotic susceptibility of all transconjugants following plasmid transfer in this study is highly reasonable. I would like to clarify that, since the *E. hormaechei* strain used in this study is a clinical isolate from a poultry farm, we selected a veterinary-specific antimicrobial susceptibility testing card, which includes most classes of antibiotics. Additionally, our supplementary analysis of another plasmid carried by *E. hormaechei* GS32 revealed that it lacks a complete conjugation transfer module, thereby excluding the possibility of inaccuracy in PCR-based validation of plasmid transfer in this study. These findings support the reliability of our results. Unfortunately, the authors of this study are currently unable to retrieve the relevant experimental materials from the original institution (Address 3). We sincerely apologize for being unable to test additional antibiotic susceptibilities to further enrich the data presented in this study. Improper statements about the transfer of antibiotic resistance in the full manuscript have been revised.

7.The Discussion is too long.

Response:

We thank the reviewer's comment. We have deleted the redundant statements in the discussion section of the manuscript and clarified the overall logic of the statement.

Re: Spectrum01979-25R1 (Potential Dissemination of IncHI2/IncHI2A Plasmids Carrying *mcr-9.4* Complex Transposon in Chicken-derived *Enterobacter Hormaechei*)

Dear Prof. Dongan Cui:

Your manuscript has been accepted, and I am forwarding it to the ASM production staff for publication. Your paper will first be checked to make sure all elements meet the technical requirements. ASM staff will contact you if anything needs to be revised before copyediting and production can begin. Otherwise, you will be notified when your proofs are ready to be viewed.

Sincerely,
Catherine Logue
Editor
Microbiology Spectrum

Reviewer #1 (Comments for the Author):

The author has addressed my concerns.